# Population Dynamics in China's Urbanizing Megaregion: A Township-Level Analysis of the Beijing–Tianjin–Hebei Region

**Yanxi Wang** [1,2,3], **Yunxia Zhuo** [1,2,3] 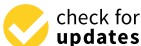 **and Tao Liu** [1,2,3,*] 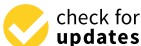

1    College of Urban and Environmental Sciences, Peking University, Beijing 100871, China
2    Center for Urban Future Research, Peking University, Beijing 100871, China
3    Key Laboratory of Territorial Spatial Planning and Development-Protection,
      Ministry of Natural Resources of China, Beijing 100871, China
\*    Correspondence: liutao@pku.edu.cn

**Abstract:** China is currently in a period of accelerated urbanization, and the population pattern of urbanizing megaregions is undergoing drastic changes. Accurately grasping the population density patterns and evolution trends has become essential. Based on the township-level population data, through population density classification, population concentration index, and regression analysis, this research investigated the evolution of the spatial pattern of population density and the influencing factors in the Beijing–Tianjin–Hebei region. Results showed that the population continued to concentrate in the municipal districts of Beijing and Tianjin and the township units where county governments were located, thereby causing a more unbalanced population distribution and a wider urban–rural disparity. Population dynamics are influenced by the market and the government. County-level administrative centers have continued to appeal to the population. The strategy of decentralizing the non-capital functions of Beijing has promoted the decentraliztion of population, albeit to a limited extent. However, key township policy has played a minor role in population change. Owing to particularities in the development stage and social system, the population dynamics in the Beijing–Tianjin–Hebei region differ from those of other developed countries.

**Keywords:** population density; pattern; influencing factor; township-level; urban–rural integration; Beijing–Tianjin–Hebei region



## 1. Introduction

Population is a carrier of economic activity [1] and a critical factor affecting regional socioeconomic development [2,3]. Changes in population distribution and influencing factors have always been an important issue in population research. Science Clark proposed the urban population density model [4], and a great deal of literature on the structure of urban population density exists. Single- [4] and multicenter structure theories [5,6] have been proposed. Additionally, urban structure evolution has drawn much attention, such as suburbanization, cyclical urbanization models [7], and shrinking city models [8]. Single- and multicenter structures also exist in regions, and the multicenter structure is the result of the strengthening of the existing links between multiple cities [9,10]. Core and periphery theory has often been used to explain the spatial distribution characteristics of economic and demographic factors. Among several regions, for various reasons, individual regions have taken the lead in developing into the core, whereas other regions have become the periphery due to slow development [11].

Regional population change can be divided into two parts: natural increase and migration [12]. Natural increase is directly affected by fertility and mortality. Population structure and other factors affect natural increase by affecting birth and death populations [13,14]. With developments in the economy and decreases in birth rate, migration has gradually become a decisive factor in regional population change [15]. Population migration theory has

become relatively mature. Scholars usually use the push–pull model to explain population migration [16,17]. When we mention the push force, harsh climatic conditions [18], limited employment opportunities, and poor living environments promote population migration to places perceived as better or more desirable. Relatively, destinations with the pull force usually have comfortable environments, sufficient job opportunities, better education and medical care, convenient transportation, good location and favorable policies. As a result, differences in natural conditions [19], industrial development [20], public services [21], traffic conditions [22], locations, and administrative factors [23] play an important role in population dynamics.

In general, most existing articles have focused on urban cases in developed countries, such as countries in Europe and North America. Studies paying attention to cities in developing countries like China are limited. The evolution of regional population distribution and their influencing factors calls for comparative research in different backgrounds. Unlike developed countries, which have entered deindustrialization and give the appearance of deurbanization, China is still in the process of rapid industrialization and urbanization. Additionally, as a socialist country, China's population dynamics are influenced by both the market and the government to a large extent. Given the difference in development stage and social systems, China's regional population changes may show different characteristics and may be affected by different factors. However, only a few studies have focused on regional population dynamics in China. Due to the limitation of data, theses studies were mostly before 2010 and on the scale of counties or cities, and were relatively coarse. An up-to-date and detailed study on China's regional population changes has not yet been done. In order to lessen this dearth of research, this paper studied the population change and influencing factors during 2000–2017 at the township level in a typical urbanizing megaregion of China, the Beijing–Tianjin–Hebei region.

Beijing–Tianjin–Hebei region was selected as a case study in this research for three reasons. First, this region has undergone a considerable urbanization process over the past few decades, in step with the whole country. The urbanization rate of the region at the time of study was 69%, close to that of China, at 64%. Second, the complex terrain, varying types of urban areas, and obvious spatial differences of the population make this region an ideal case study with great potential reference significance to population dynamics in urbanizing megaregions. The Beijing–Tianjin–Hebei region has formed a multicenter structure centered on Beijing, Tianjin, and Shijiazhuang [24]. Moreover, the population density change in the region presented the trend of centralization and imbalance [24,25]. The population proportion of Beijing and Tianjin increased continuously, and the characteristics of the population density circle grew prominent [26]. Third, as the capital circle of China, the Beijing–Tianjin–Hebei region has special national strategic significance. The national strategies of decentralizing the non-capital functions of Beijing and promoting the coordinated development of the Beijing–Tianjin–Hebei region were launched in 2015 [27]. Since then, many non-capital functions, such as the manufacturing industry, regional logistics base, and wholesale market, have been moved from Beijing to Tianjin and Hebei to promote the decentralization of the population [28]. Exploring the population dynamics of this region and the effect of the strategy will provide significant references for other regions.

In this paper, we studied the population distribution evolution and influencing factors of the Beijing–Tianjin–Hebei region during 2000–2017 at the township level, attempting to answer the following questions and make contributions to the literature. First: has the trend of population concentration to Beijing, Tianjin, and other central urban areas changed? Most studies on population distribution and change in the Beijing–Tianjin–Hebei region were based on census data available in and before 2010 [25,29]. They revealed the imbalanced trend of population distribution in Beijing–Tianjin–Hebei region [30]. However, with the implementation of the Beijing-Tianjin-Hebei coordinated development strategy, the geographical distribution of the population might have changed in recent years. The decentralization of non-capital functions from Beijing to Tianjin and Hebei may have



led to a more balanced population in the region. Second: what are the characteristics of population distribution at the county scale based on the township-level data? Research on population density pattern is scale-dependent [31,32]. With the expansion of spatial scale, spatial heterogeneity was reduced, and population distribution characteristics in the region were covered [33]. Research at the county level ignore the distribution and spatial changes of population within counties. Under the background of rapid urbanization in China, the socioeconomic space within counties has been undergoing dramatic change, significantly impacting the spatial population distribution [34,35]. Therefore, exploring the population distribution at the township level was deemed urgent. Third: what are the differences in trends and influencing factors of population evolution between the Beijing–Tianjin–Hebei region and metropolitan areas in developed countries? Affected by deindustrialization, many regions in developed countries have shown a trend toward suburbanization and deurbanization [36–38]. Given the socialist system, the population change in the Beijing–Tianjin–Hebei region was not only affected by market factors but also constrained by planning and related policies. The differences in population dynamics between the Beijing–Tianjin–Hebei region and metropolitan areas of capitalist countries were considered worth analyzing.

The remainder of the paper was organized as follows: Section 2 introduces the study area, method, and data used in this research. Section 3 presents the results. Section 4 reports the discussions. Section 5 provides the conclusions.

## 2. Data and Methods

### 2.1. Study Area

The study area was the Beijing–Tianjin–Hebei region, located in the core area around the Bohai Sea in northeast Asia and China (Figure 1). The region has diverse topographic conditions, including resources such as mountains, plains, rivers, and lakes. The northwest of the Beijing–Tianjin–Hebei region is a mountainous area with a high slope, and the east has gradually transformed into a plain area. Although it covers only 2.25% of Chinese territory (9.6 million km$^2$), this region contained 8.09% of the national population (1.39 billion), and it produced 9.98% of the total national gross domestic product (12,250.3 billion dollars) in 2017. At present, the interconnected comprehensive transportation network in the Beijing–Tianjin–Hebei region has basically taken shape, and the comprehensive transportation service level has continued to improve.

The Beijing–Tianjin–Hebei region includes two municipalities and 11 prefecture-level cities. At the county level, the region can be divided into municipal districts, county-level cities, and counties. The region comprised 79 municipal districts, 20 county-level cities, and 101 counties in 2017. Although the units of three administrative types belonged to the same administrative level, some differences in economic form and industrial structure were observed among them [39]. Municipal districts are usually located in the central areas of the cities and cooperate with other districts to realize the leading function of the cities. Their industry and commerce were developed to a mature level, and nonagricultural industries occupied a dominant position. Most counties had prominent agricultural functions, and the industrial and commercial development level was still in the primary stage, i.e., they could be regarded as rural areas. County-level cities referred to those in a transitional state between counties and municipal districts. Their industrial and commercial development was better than that of counties and weaker than that of municipal districts. In terms of the realization of the functional mechanism, municipal districts realized the functions of cities through cooperation with other municipal districts, whereas county-level cities and counties performed their functions independently. Therefore, some differences were found in the population distribution and change within the three types of administrative departments.

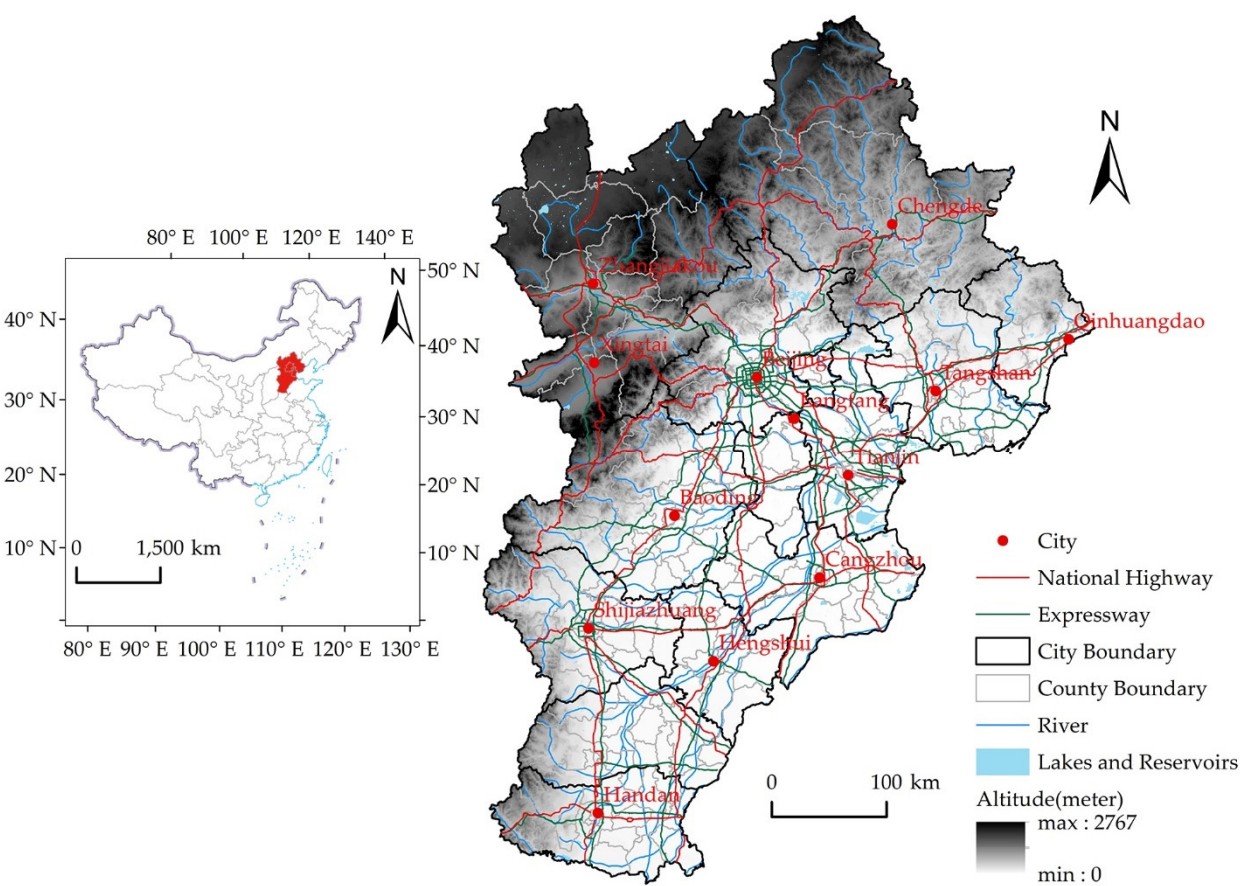

**Figure 1.** Research area.

*2.2. Methods*

2.2.1. Population Concentration Index

The population concentration index is a quantitative index to judge the concentration of regional population distribution [40], and its calculation formula is as follows:

$$C = \frac{1}{2} \sum_{i=1}^{n} |x_i - y_i|, \tag{1}$$

where $C$ is the population concentration index; $n$ is the number of research units; $x_i$ is the proportion of area $i$ population in the total population of the study area; $y_i$ is the proportion of the land area of area $i$ in the total land area of the study area. The smaller the value of $C$, the more balanced the population distribution; on the contrary, the more concentrated the population distribution.

2.2.2. Population Density Classification

Population density is the most important form of population distribution and the main indicator to measure regional differences in population distribution [29]. The fine classification of population density can lead to useful information, but too much classification causes a certain amount of information redundancy and covers up some distribution laws. Therefore, this study made reasonable adjustments to the quantile classification method, obtained appropriate classification results, and performed statistical analyses on the quantity and cumulative value of each grade.

2.2.3. Model Specification

(1) Model

The regression model was constructed as follows:

$$\text{den} = \beta_0 + \beta_1 \text{Ind} + \beta_2 \text{Nat} + \beta_3 \text{Ser} + \beta_4 \text{Tra} + \beta_5 \text{Loc} + \beta_6 \text{Adm} + \varepsilon, \tag{2}$$

$$\Delta \text{den} = \beta_0 + \beta_1 \text{Ind} + \beta_2 \text{Nat} + \beta_3 \text{Ser} + \beta_4 \text{Tra} + \beta_5 \text{Loc} + \beta_6 \text{Adm} + \beta_7 \text{den2010} + \varepsilon, \tag{3}$$

where den and $\Delta$den are the dependent variables that indicate the population density change and population size of each unit, respectively. Ind, Nat, Ser, Tra, Loc, Adm, and den2010, respectively, represent industrial development, natural factor, public service, traffic condition, location factor, administrative factor, and base population density. $\beta_1$, $\beta_2$, and $\beta_n$ represent the coefficients of independent variables, and $\varepsilon$ is the residual term. The ordinary least squares (OLS) method was used to estimate the parameters.

(2) Independent variable

This study took population distribution and change as research objects and correspondingly took population density in 2017 and population density changes from 2010 to 2017 as indicators. Population density changes are composed of natural and mechanical growths, but the impact of natural growth has been gradually weakening. Migration was observed to have an increasing impact on regional population change. This research mainly analyzed regional population change from the population migration perspective. Previous studies have revealed that population migration was influenced by various factors, such as nature [19,41], economy [42,43] and institution [23,44]. On the basis of the literature, and considering the availability of township-level data, we selected 16 variables in 7 categories, including initial population density, natural factor, industrial development, public service, traffic condition, location, and administrative factor.

A. Initial population density

Due to the effects of economies of scale and agglomeration economies [45], with the expansion of unit size, production efficiency increases to attract population agglomeration [46]. However, an extremely large population may lead to the decrease of urban efficiency due to the crowding effect [47]. We used the population density of each unit in 2010 to characterize the initial size of each unit and explore the impacts of initial population size on population change in the Beijing–Tianjin–Hebei region.

B. Natural factor

In terms of natural factors, average altitude and average slope are selected as influencing factors. Altitude is an essential indicator of human comfort [48]. Slope factor mainly affects the population pattern by affecting the costs and conveniences of various types of human construction [49]. Population distribution in areas with high altitudes and large slopes is relatively sparse and has a negative impact on population growth.

C. Industrial development

Differences exist in the industrialization and urbanization levels of different township units. For units in the early stages of industrialization, industry can attract labor [50]. We chose the density of industrial points of interest (POI) as an indicator to measure the industrial development level of a unit. The population densities of units with dense industrial layouts were expected to be high, and populations likely to show growth trends.

D. Public service

In addition to obtaining improved employment opportunities, high wages, and high expected wages, laborers also migrate for good public services [51–53]. We used the densities of POIs in primary and secondary schools and hospitals to measure the supply of regional public services. We hypothesized that good public service supply would have a positive impact on population size and population change.

E. Traffic condition

The impact of traffic on regional population works mainly to reduce the travel costs of residents to affect population concentration and decentration [54]. The optimization of urban local traffic conditions leads to a decrease in transport costs, thereby enhancing the

accessibility of cities or regions [55], which is conducive to urban or regional population concentration [56]. In this study, we selected the densities of provincial highways, county highways, and urban trunk roads to represent the local traffic conditions of township units. We also used the densities of railways, expressways, and national highways to represent the regional traffic conditions of township units. We expected that, the higher the road density, the better the traffic conditions, the greater the population density, and the faster the population growth.

F. Location factor

Administrative centers in counties, cities, and the capital can provide different levels of public services, and township units close to these areas can have good development opportunities, resulting in high population density and fast population growth [57,58]. Considering that the impacts of different administrative regions may be different, two kinds of indicators were used to characterize the nearest administrative center distance and the administrative center distance.

G. Administrative factor

Some township units have special political attributes in the Beijing–Tianjin–Hebei region. In 2014, China released a list of national key townships. National key townships generally have the characteristics of large town size, large population, relatively developed economy, and relatively complete supporting facilities. The construction of national key towns is an effective means to balance the distribution of rural transferred population and reduce the pressure on the population in large cities. Provincial key townships are the regional nodes of key development selected in the urban system planning of each city and usually have a good industrial economic foundation. Compared with other township units, they have a better economic development level and generally become the regional development pole. We divided such units into three types of regions, among which national or provincial key township units had a solid industrial and commercial foundation and great development potential in the future. Compared with major cities, they had lower access thresholds and lower living and social costs. They were also more prone to attract population aggregation to achieve their own growth. In addition to having more commercial facilities, township units where city and county governments were located usually concentrated local government departments. At the same time, they were able to provide strong public services and had a strong attraction to the population. These variables are described in Table 1 below.

**Table 1.** Selection and descriptions of township-level indicators.

| Category | Variables | Unit | Mean | SD | Min | Max |
|---|---|---|---|---|---|---|
| Dependent variable | | | | | | |
| | Population density change | people per km$^2$ | 84.97 | 735.22 | −7868.26 | 14,690.24 |
| | Population density in 2017 | people per km$^2$ | 925.71 | 2463.86 | 3.26 | 34,779.84 |
| Independent variable | | | | | | |
| Initial population density | Population density in 2010 | people per km$^2$ | 840.74 | 2057.55 | 2.40 | 27,741.10 |
| Natural factor | Average altitude | m | 281.48 | 416.08 | 0.49 | 1698.22 |
| | Average slope | degree | 1.59 | 2.32 | 0 | 11.11 |
| Industrial development | Density of POI in industry | points per km$^2$ | 0.02 | 0.10 | 0 | 2.19 |
| Public service | Density of POI in hospital | points per km$^2$ | 0.13 | 0.58 | 0 | 12.89 |
| | Density of POI in primary and secondary school | points per km$^2$ | 0.11 | 0.29 | 0 | 7.14 |

**Table 1.** *Cont.*

| Category | Variables | Unit | Mean | SD | Min | Max |
|---|---|---|---|---|---|---|
| Traffic condition | Local road density | km per km$^2$ | 0.18 | 0.28 | 0 | 2.98 |
| | Regional road density | km per km$^2$ | 0.72 | 1.03 | 0 | 13.45 |
| Location factor | Distance to the administrative center of the county | km | 17.23 | 13.42 | 0 | 89.65 |
| | Distance to the nearest county administrative center | km | 14.47 | 10.15 | 0 | 72.02 |
| | Distance to the administrative center of the city | km | 73.42 | 78.42 | 0 | 376.71 |
| | Distance to the nearest city administrative center | km | 50.04 | 26.30 | 0 | 174.86 |
| | Distance to Beijing administrative center | km | 201.73 | 104.72 | 0 | 455.93 |
| Administrative factor | National key township | — | 0.10 | 0.30 | 0 | 1 |
| | Provincial key township | — | 0.05 | 0.22 | 0 | 1 |
| | District and county administrative center unit | — | 0 | 0.06 | 0 | 1 |

*2.3. Data*

In this study, permanent population data at the township level in 2000 and 2010 were obtained from the fifth and sixth national censuses, respectively. Permanent population data at the township level in 2017 were taken from the China County Statistical Yearbook (Township Volume) 2018. Permanent population data of districts and counties in the Beijing–Tianjin–Hebei region in 2017 were derived from the Beijing Statistical Yearbook 2018, the Tianjin Statistical Yearbook 2018, and the Hebei Economic Yearbook 2018. In addition, we used the 2018 statistical yearbooks of districts and counties in Hebei Province for verification and interpolation to ensure data integrity and accuracy. Considering the availability of data and the necessity of new trends in the region, we selected research periods in 2000, 2010, and 2017. To maintain the continuity and comparability of the research units, the divisions of 2000 and 2010 were adjusted to be consistent with the administrative divisions of 2017; hence, the data of the three years were matched in space. Subdistricts in the same county unit are continuous in the built-up area, with similar socioeconomic conditions, so they could also be regarded as a whole. We also merged the subdistricts within a county (district) into a separate unit. On the contrary, each township unit was independent. In addition, since 2000, the administrative divisions of township units changed frequently, including the change of townships to subdistricts and the internal adjustment of townships. We merged and split the corresponding units according to the administrative division adjustment documents (http://www.xzqh.org/html/, accessed on 1 January 2021) and finally formed a township-level permanent population dataset for three years. The dataset included 2312 units of three types: townships, towns, and subdistricts, all of which were called townships below.

Elevation data and average slope data used the national DEM 1:1 million data from the geographic data platform of Peking University (https://geodata.pku.edu.cn/, accessed on 1 January 2021). The road data and POI data were from the 2012 Auto Navi Map national POI database. The list of national key township units was from the Ministry of Housing and Urban–Rural Development (https://www.mohurd.gov.cn/gongkai/fdzdgknr/tzgg/2014 07/20140731_218612.html, accessed on 1 January 2021). The list of provincial key township units was from the urban planning departments of provinces and cities. The number of POIs in each township unit could be obtained using the ArcGIS Software Zonal statistics tool in ArcGIS software.

## 3. Results

### 3.1. Population Distribution Evolution

Population density characteristics were similar in the selected three years. To understand the general population distribution in the three years, the logarithm of the population density of each township unit in 2000, 2010, and 2017 was taken and the kernel density as estimated, as illustrated in Figure 2. The results are close to normal distribution, and the results of the three years are similar in morphology. The probability density function at $x \approx 6.3$ (corresponding to the population density of about 544 people per km$^2$) reached the peak, and the probability density decreased gradually from 2000 to 2017 at the peak. Therefore, the proportion of township units with medium population densities was gradually decreasing, and that of township units with high and low population densities was increasing.

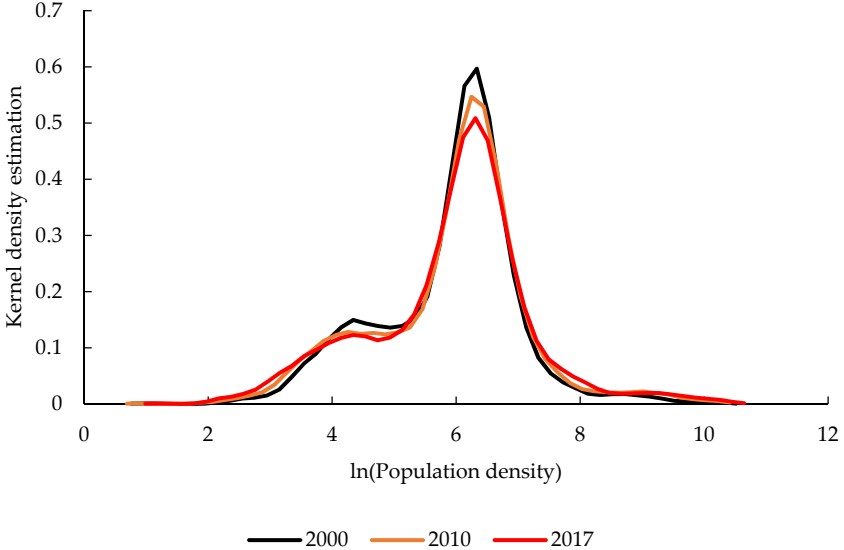

**Figure 2.** Log kernel density estimation of population density in the Beijing–Tianjin–Hebei region from 2000 to 2017.

The population in the Beijing–Tianjin–Hebei region was still gathering at the time of our study, but the gathering speed slowed down slightly. According to the population concentration index in Table 2, the indexes in 2000, 2010, and 2017 were 0.442, 0.476, and 0.501, respectively. The index increased year by year, but the growth rate gradually slowed down. The population concentration in the Beijing–Tianjin–Hebei region showed an upward trend, but the population concentration speed slowed down. To explore the differences among different units in this area, we calculated the range and variance of population density in three years at the township level (Table 2). The results showed that the variance of township unit population density in the Beijing–Tianjin–Hebei region continued to expand. According to the differences in population densities between urban and rural areas, units with high population densities could be regarded as urban units, and units with low population densities could usually be regarded as rural units. At the township level, the imbalanced development in this region was expanding, which also implied that the urban–rural difference in the Beijing–Tianjin–Hebei region continued to expand. The proportion of units with high population densities continued to increase, and that of township units with medium population densities decreased, suggesting that the regional population agglomeration made the areas with high economic levels gather more population than other areas.

**Table 2.** Population distribution indexes in the Beijing–Tianjin–Hebei region.

| Year | 2000 | 2010 | 2017 |
|---|---|---|---|
| Population concentration index | 0.442 | 0.476 | 0.501 |
| Population density variance | 2,851,766 | 4,233,521 | 6,070,625 |

The spatial distribution pattern of population density was relatively stable, and a certain East–Middle–West strip feature existed (Figure 3). The northwest mountainous area was a low-density population area, except the township units where county governments were located, indicating that the elevation factor had a negative impact on population distribution. The central Beijing–Baoding–Shijiazhuang–Xingtai–Handan formed a high-density population concentration zone. In the eastern region, except Tianjin, most areas belonged to the medium density population belt. In the area, high- and low-density population belts were staggered. In the three years of 2000, 2010, and 2017, population differentiation characteristics in the Beijing–Tianjin–Hebei region slightly changed, suggesting that the macro pattern of population density distribution had a certain stability.

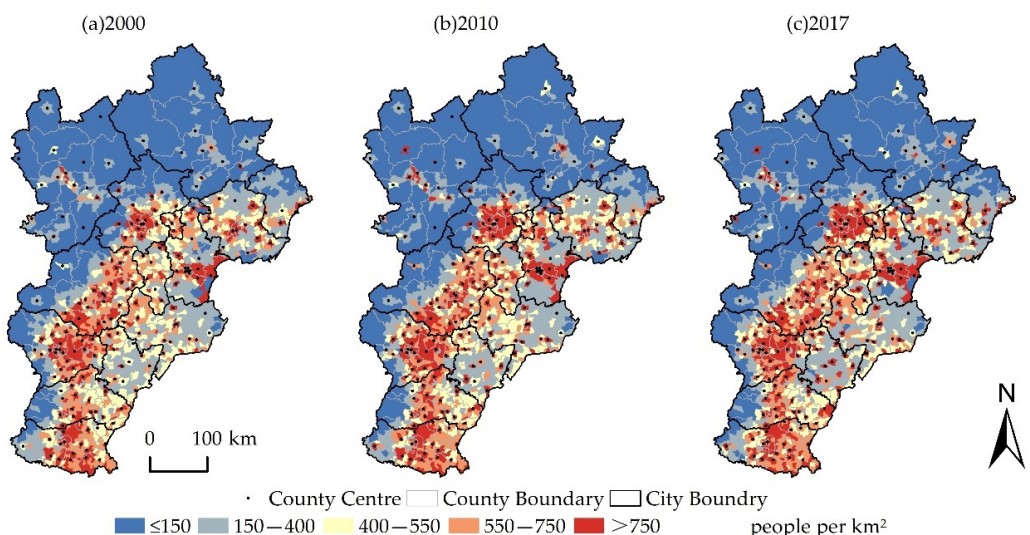

**Figure 3.** Distribution pattern of population density in the Beijing–Tianjin–Hebei region from 2000 to 2017.

Within the district and county units, the population distribution had obvious center-periphery characteristics (Figure 4). In each county unit, the township unit where the county government was located was an area with high population density. The farther away from the center, the more the population density gradually decreased. This phenomenon was obvious in the low-density population belt in the east. Township units where district and county governments were located, for the most part, had good economic development levels and were able to provide good public services. Thus, such units easily attracted population centralization. The farther away from the center, the more difficult it was for township units to enjoy the services provided by the center. Thus, the attraction to the population gradually weakened, and the population density gradually decreased.

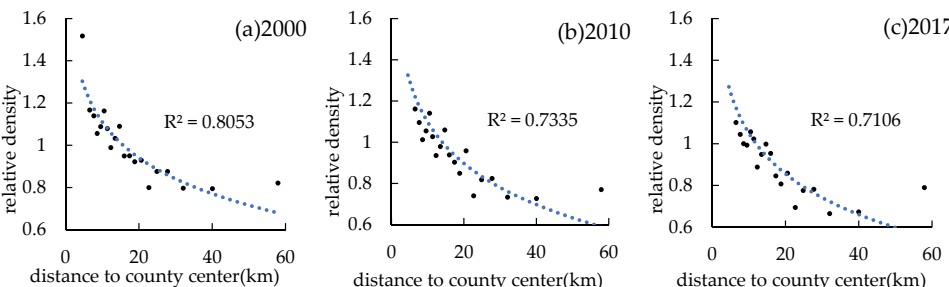

**Figure 4.** Binscatter diagram of the relative density of distance from each township unit to the administrative center of a county.

*3.2. Spatial Characteristics of Population Increase and Decrease*

The population continued to gather in municipal districts and township units where district and county governments were located. The proportion of units with increasing population during the 2000–2010 period was 53.37%, slightly higher than the number of units with decreasing population. Figure 5 illustrates that population density changes in most township units were concentrated in a certain range (from −50 to 50 people per km$^2$), and the proportion of significant increase levels (>100 people per km$^2$) in township population density in municipal districts was higher than that in county and county-level cities. Therefore, municipal districts were the main areas for population centralization. Areas with the fastest population density increase levels were mainly concentrated in the central urban areas of Beijing and Tianjin, including township units where county governments were located. Population density increases in township units where district and county governments were located were relatively high, and population intensity changes in surrounding areas were slightly reduced, also showing a certain center-periphery structure. Units with population loss were mainly concentrated in northwest high-altitude areas, but population loss was not serious, generally ranking at the slight loss level (from −50 to 0 people per km$^2$). The population loss of township units in some urban fringe areas was serious. For example, the population loss intensity values in the urban fringe areas of Hengshui, Shijiazhuang, and Handan were relatively high (Figure 5).

The share of units with population increase was 53.20% from 2010 to 2017, similar to the previous phase, but the spatial distribution and intensity of population density changes transformed dramatically. Several township units were still concentrated at the level with population changing slightly (from −50 to 50 people per km$^2$). The proportion of township units with large population loss (≤−100 people per km$^2$) showed an increasing trend, whereas that of units with large population increase (>100 people per km$^2$) decreased. Compared with the 2000–2010 period, the proportion of units with large population loss (≤−100 people per km$^2$) in municipal districts significantly increased, indicating that the trend of population change within municipal districts transformed from 2010 to 2017. Units with large population increases were initially located in Beijing and Tianjin, and then transformed into the periphery of Beijing and the central urban area of Tianjin. In addition, township units where the administrative centers of districts and counties were located remained in the concentrated areas of population. In high-altitude areas, some units with slight population loss (from −50 to 0 people per km$^2$) in the past transformed into slight population increases (from 0 to 50 people per km$^2$). Township units where county administrative centers were located remained in the main area of population agglomeration. Population loss in the eastern and central plains were increasing. Urban fringe areas in Hengshui, Handan, Xingtai, and other cities showed different degrees of population loss, and population loss intensities in urban fringe areas were relatively high (Figure 5).

Municipal districts were still the main gathering places for population agglomeration. We calculate the number of township units with different levels of population density changes (Figure 6a). From 2000 to 2010, most areas with population loss were the units in counties. Units with large population increases (>100 people per km$^2$) were mainly

located in municipal districts. The population changes in this period showed that the population in the Beijing–Tianjin–Hebei region was gathering in municipal districts, and populations in some units in counties were decreasing slightly. From 2010 to 2017, the number of units with large population losses ($\leq-100$ people per km$^2$) increased, whereas that of units in municipal districts with large population increases (>100 people per km$^2$) decreased. Therefore, the population in the Beijing–Tianjin–Hebei region began to be highly concentrated in specific municipal districts, and the attraction of these municipal districts to the population further expanded.

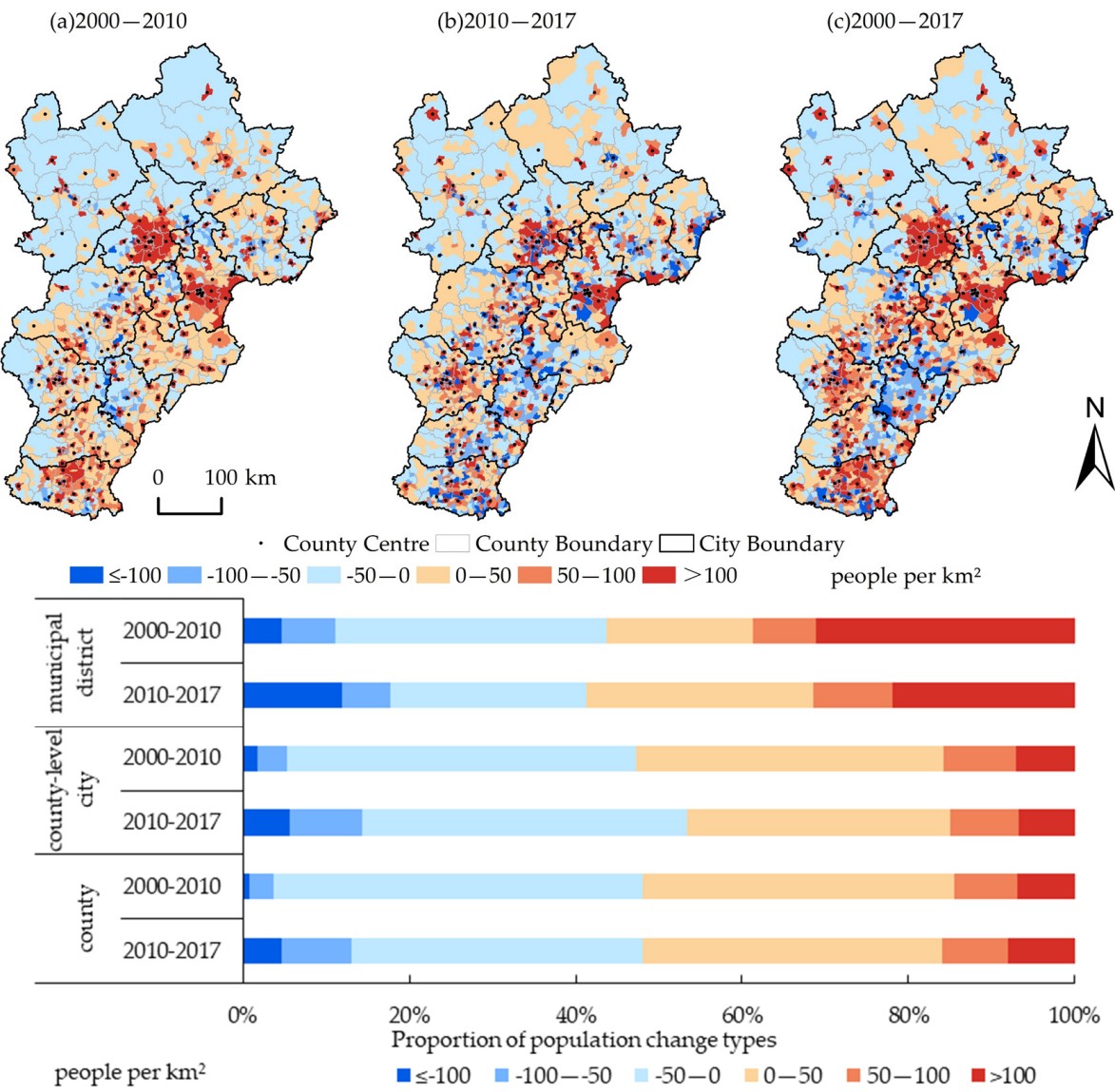

**Figure 5.** Spatial distribution of population density changes in the Beijing–Tianjin–Hebei region from 2000 to 2017 and the proportion of various types.

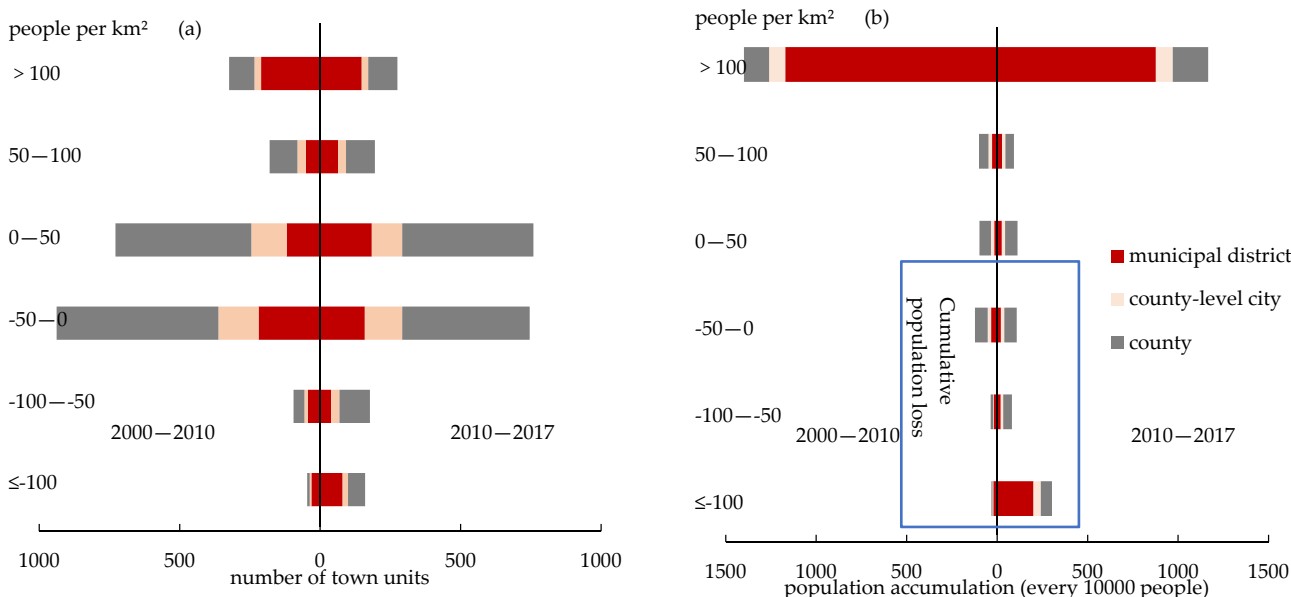

**Figure 6.** (**a**) Number of units at each level of population density change; (**b**) Population accumulation of population density change at all levels.

To further explore the specific situations of population changes in the three types of areas, we accumulated the population change of each unit, according to each level, to form Figure 6b. From 2000 to 2010, the cumulative population values of urban units with large population increase levels (>100 people per km$^2$) dominated, suggesting that a large number of people gathered in urban units at this stage, and the population loss was mainly located in some rural units. From 2010 to 2017, the basic pattern slightly changed, and the increase of urban and rural units with large population losses led to the increase of the overall population loss. With further city development, some urban units began to lose population.

### 3.3. Population Change Type Analysis

To understand, in detail, the transition of population changes in different township units in the two periods of 2000–2010 and 2010–2017, according to the positive and negative growth of each township unit in the two stages, township units were divided into four types: continuous increase, from decrease to increase, from increase to decrease, and continuous decrease (Figure 7).

Units of continuous increase were in the state of population increase in the two periods, sustaining attraction to the population. Such township units accounted for 30.14% of the region and were mainly distributed within municipal districts. These units were mainly concentrated in the eastern and central areas of the Beijing–Tianjin–Hebei region (Figure 7). The Beijing–Tianjin, Tianjin–Shijiazhuang, and Shijiazhuang–Handan belts were the main areas where traffic lines were concentrated. Traffic corridors improved regional traffic accessibility, reduced traffic costs, and proved able to attract many people. Convenient transportation facilities can improve enterprise accessibility to the market. At the same time, high population mobility makes it possible for enterprises to obtain high benefits. Therefore, enterprises prefer such areas, and production and population begin to concentrate in these areas. The units of sustained growth in the northwestern region were usually the township units where district and county governments were located. Such regions have high population densities, which can promote the matching of the employment market and the full release of consumption capacity. At the same time, they have relatively perfect urban infrastructure and supporting services, which can continuously attract population.

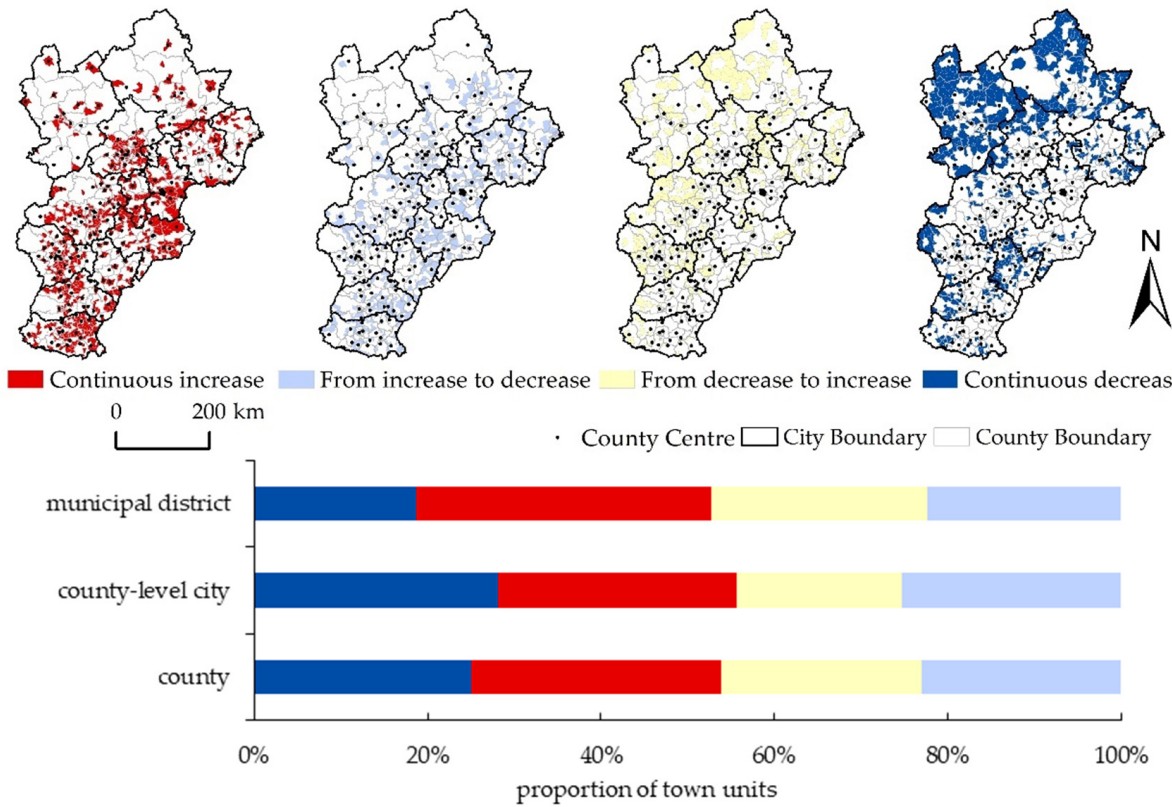

**Figure 7.** Types of population changes and the proportion of various types.

Units of the increase to decrease type referred to areas where population changed from agglomeration to loss, accounting for 23.2%. Such areas had relatively high distribution proportions at the county level, and population loss in non-urban center areas was likely to occur. They were mostly distributed in urban fringe areas, and some areas are township units where county governments were located. The factors that weakened the population agglomeration ability were diversified. As an example, the weakening of such an ability of some township units in Beijing may have been due to the relocation of high-energy-consuming industries caused by the policy to relieve Beijing of functions nonessential to its role as China's capital (http://www.beijing.gov.cn/renwen/bjgk/jjj/ghgy/202007/t2 02007231956512.html, accessed on 1 January 2021), thus leading to the phenomenon of population loss. Most other township units could lead to a decline in population with the transfer of labor-intensive industries, due to factors such as rising costs, industrial transfers, and industrial restructuring.

Units of the decrease to increase type referred to the transition of population from loss to growth, which accounted for 23.0%. Such areas were also mainly distributed within municipal districts (Figure 7), suggesting that the populations of township units in urban areas were likely to increase. Such areas were usually distributed in the surrounding areas of township units where county governments were located. Population growth in such areas was small, but with the improvement of transportation conditions within the region, transportation costs gradually decreased. Therefore, they were able to rely on low costs and other advantages to achieve rapid economic development by undertaking industries in the central urban area to enhance the population agglomeration ability.

Units of continuous decrease referred to areas that are always in the state of population loss, accounting for 23.7%. Such areas were mainly located within county-level cities and counties. They were distributed in patches in the northwestern region. When we focused on the city level, these areas were usually distributed around the center of high-density populations. In general, the economic development of such areas was relatively poor, and

the core areas of the region absorbed additional resources and populations from these places due to the siphon effect.

### 3.4. Influencing Factors of Population Distribution and Change

Based on the above analysis, the population distribution and changes at the township level were affected by diversified factors, such as nature, economy, industry, transportation, location, and administration. In this study, OLS regression was used to explore the influences of specific factors. Many independent variables were involved; thus, multicollinearity could be observed among independent variables. A variance inflation factor (VIF) test was performed on the model to detect the multicollinearity of factors. The results showed that the VIF value of each factor as lower than 6, and the multicollinearity of factors was weak. The regression results are presented in Table 3.

The above regression results indicated that the population density distribution of each unit in 2017 was related to natural factors, industrial development, public service, traffic condition, location factors, and administrative factors. Population density changes were related to other factors, except natural factors. The population density change variable was negatively correlated with the population density variable, suggesting that units with high-density populations showed a downward trend from 2010 to 2017. Combined with the above analysis of population density change, the effects of the policy to relieve Beijing of functions nonessential to its role as China's capital were obvious.

The impacts of natural factors on population density in 2017 were remarkable. The population density decreased as altitude and slope increased, indicating that natural factors determined the basic pattern of population distribution. However, the limitation of natural factors on population density change was weak.

From the regression results, the population density variable in the Beijing–Tianjin–Hebei region showed a significant negative correlation with the density levels of industrial sites, suggesting that labor force absorption by industrial development was low. Therefore, the corresponding population density was low in areas with the dense layout of industrial facilities. Population density change was also related to the layout of industrial facilities. In the units of non-administrative centers, the densities of industrial enterprises had significant negative effects on population density growth, indicating that the development model of heavy industry in regional development may not promote population growth in ordinary units.

The layout of public facilities affects the supply of unit public services, and the density layout of public facilities represents the supply capacity of unit public services. The regression results revealed that hospital variables were positively correlated with population density variables and population density change variables. Thus, the population distribution matched medical services, and the unit population with good medical services exhibited significant growth. The primary and secondary school density variables showed a significant positive correlation with the population distribution of ordinary town units, suggesting a matching relationship between educational services and population density. When all units participated in the regression, the performance of this variable was insignificant because the scale of the school was relatively large in units where district and county administrative centers were located. Moreover, the existing POI data was unable to reflect this situation. With the participation of all units, this variable was negatively correlated with the population density variable, which was also the common reason.

**Table 3.** Population distribution and change regression results from 2010 to 2017.

| | | (1) | (2) | (3) | (4) | (5) | (6) | (7) | (8) |
|---|---|---|---|---|---|---|---|---|---|
| | **Variables** | **Population Density in 2017** | | | | **Population Density Change from 2010 to 2017** | | | |
| Initial population density | Population density in 2010 | | | | | −0.113 *** (−5.96) | −0.113 *** (−5.96) | −0.053 *** (−2.99) | −0.053 *** (−2.98) |
| Natural factors | Average altitude | −0.089 * (−1.75) | −0.115 ** (−2.39) | −0.023 (−0.32) | −0.029 (−0.43) | 0.014 (0.40) | 0.011 (0.32) | −0.003 (−0.05) | −0.006 (−0.14) |
| | Average slope | −27.208 *** (−3.20) | −27.586 *** (−3.25) | −25.077 ** (−2.05) | −24.687 ** (−2.02) | 1.105 (0.18) | 0.996 (0.17) | −1.607 (−0.20) | −1.567 (−0.19) |
| Industrial development | Density of POI in industry | −1311.145 *** (−4.01) | −1318.731 *** (−4.03) | −2176.037 *** (−7.15) | −2191.638 *** (−7.20) | −2108.716 *** (−9.20) | −2106.823 *** (−9.19) | 161.628 (0.77) | 161.626 (0.77) |
| Public service | Density of POI in hospital | 2987.775 *** (26.93) | 2966.548 *** (26.90) | 2781.022 *** (26.52) | 2777.781 *** (26.64) | 881.672 *** (9.83) | 879.658 *** (9.88) | 853.804 *** (10.77) | 852.302 *** (10.80) |
| | Density of POI in primary and secondary school | 486.081 *** (2.68) | 505.531 *** (2.79) | 75.480 (0.38) | 87.199 (0.44) | 103.420 (0.81) | 103.650 (0.82) | −555.231 *** (−4.16) | −554.411 *** (−4.16) |
| Traffic condition | Local road density | 284.448 *** (3.35) | 299.208 *** (3.52) | 1433.719 *** (15.04) | 1448.179 *** (15.20) | 50.390 (0.84) | 50.571 (0.85) | 287.847 *** (4.26) | 289.549 *** (4.28) |
| | Regional road density | 435.733 *** (12.09) | 445.386 *** (12.51) | 573.931 *** (13.73) | 577.554 *** (13.94) | 153.619 *** (5.93) | 154.565 *** (6.02) | 28.134 (0.94) | 29.468 (0.99) |
| | Distance to the administrative center of the county | | 0.288 (0.23) | | 3.795 ** (2.08) | | −0.081 (−0.09) | | 0.869 (0.71) |
| | Distance to the administrative center of the city | | −0.145 (−0.76) | | −0.020 (−0.08) | | −0.068 (−0.50) | | −0.108 (−0.60) |
| Location factors | Distance to the nearest county administrative center | 0.189 (0.10) | | 6.154 ** (2.17) | | −0.260 (−0.19) | | 1.338 (0.70) | |
| | Distance to the nearest city administrative center | −1.135 * (−1.68) | | −1.171 (−1.23) | | −0.111 (−0.23) | | −0.375 (−0.59) | |
| | Distance to Beijing administrative center | 1.081 *** (7.36) | 1.097 *** (7.15) | 1.376 *** (6.74) | 1.386 *** (6.48) | 0.339 *** (3.26) | 0.352 *** (3.25) | 0.244 * (1.76) | 0.269 * (1.85) |
| | National key township | 24.527 (0.47) | 24.736 (0.47) | | | 3.198 (0.09) | 3.302 (0.09) | | |
| Administrative factors | Provincial key township | −150.547 ** (−2.25) | −153.247 ** (−2.29) | | | −10.148 (−0.22) | −10.083 (−0.22) | | |
| | District and county administrative center unit | | | 354.700 *** (4.01) | 328.217 *** (3.88) | | | 350.888 *** (5.91) | 346.338 *** (6.10) |
| | Intercept term | 27.463 (0.47) | −24.977 (−0.47) | −352.236 *** (−4.50) | −389.959 *** (−5.58) | −107.536 *** (−2.60) | −112.816 *** (−3.05) | −67.8216 (−1.28) | −79.192 * (−1.67) |
| | Number of samples | 2114 | 2114 | 2312 | 2312 | 2114 | 2114 | 2312 | 2312 |
| Adj.R² | | 0.728 | 0.727 | 0.855 | 0.855 | 0.096 | 0.096 | 0.264 | 0.265 |

*t* statistics in parentheses. * $p < 0.1$, ** $p < 0.05$, *** $p < 0.01$.

The improvement of traffic conditions helps reduce the transportation costs of units within cities and at the regional level. Local and regional road densities have significant positive effects on unit population density and population density change. Thus, the improvement of traffic conditions in town units was helpful in promoting the population growth of the areas. The location characteristics of each township unit could be characterized by the distance to the administrative centers at all levels. The distance to Beijing administrative center had a significant positive impact on the population density and population density change. The population distribution of the Beijing–Tianjin–Hebei region still belonged to the structure of Beijing as the center of high-density population. From the population change perspective, the center of high-density population in Beijing showed itself to be in a trend of outward diffusion, which also showed that the policy to relieve Beijing of functions nonessential to its role as China's capital had achieved initial results. On the basis of controlling other factors, other location factors were insignificant, suggesting that the diffusion effect of administrative centers at all levels as not obvious.

The influences of administrative factors were reflected in their own resource endowment. From the results, the provincial key towns had a significant negative impact on the total population, indicating that the population density of provincial key towns as still at a low level and needed further cultivation. In addition, township units where county governments were located had a significant positive impact on the total population, suggesting that the population as highly concentrated in townships where county governments were located. The influences of provincial and national key township units on population change were not obvious. These key towns did not show a strong state of population agglomeration. At present, the policy of national key township units has not led to a significant increase in population. It still needs some relevant policies, such as industrial enterprises and commerce and economic support, to realize the cultivation of key towns and enhance their ability to gather population. For township units where county governments were located, the population showed an aboriginal increasing trend. Hence, administrative centers had a strong positive impact on population agglomeration, and the service facilities within units were able to attract population agglomeration. Strengthening the connection between county government units and the surrounding township units is still necessary to effectively promote the development of surrounding areas.

## 4. Discussion

### 4.1. Particularity of Population Distribution and Change

First, the population distribution in the Beijing–Tianjin–Hebei region could be seen as a multicenter distribution structure, similar to North American metropolitan areas. However, some differences were found in the population distribution at the city level. Most empirical studies have shown that the urban population density gradient in North America shows characteristics of overall decline, the beginning of central depression, the decline of peak density, and the low degree of fitting around the trend gradient (with great variability) [59]. At the same time, various urban problems, such as high taxes and crisis, may occur in central cities [60]. The urban population distribution in the Beijing–Tianjin–Hebei region still followed the distribution characteristics of center-periphery: within the city, the municipal area was a high-density area of urban population distribution, far away from the urban center, and the population density gradually decreased. Units where district and county governments were located were usually areas with the highest population densities of districts and counties, and the relative population density began to decline gradually farther away from the administrative centers of districts and counties. The main reasons for the population differences between the Beijing–Tianjin–Hebei region and the metropolitan areas of North America and Europe were as follows. Different development stages: the United States and European countries are developed, the urbanization rates of their cities are at high levels; most cities in metropolitan areas have also experienced the transformation of "Urbanization–Suburbanization–Deurbanization–Reurbanization" [61]. Due to the deindustrialization and improvement of transportation facilities in metropolitan

areas in Europe and North America, people have chosen to migrate to the suburbs in pursuit of higher quality of life, resulting in the decline of the population in the central urban area, showing the characteristics of depression [37]. However, the Beijing–Tianjin–Hebei region is still in the rapid urbanization process, and the central urban areas maintain a strong attraction. The central urban areas of cities and units where district and county governments are located can provide improved infrastructure and public services, which are usually areas with relatively concentrated population distribution.

Second, different social systems lead to the differentiation of urban development modes. Countries in Europe and North America are capitalist, with obvious market-oriented characteristics in their urban development processes [62]. Industrialization had a profound impact on the urban development process. After the industrial recession, the quality of life in central urban areas decreased, the population began to move out, and central urban areas showed characteristics of population depression [8]. In socialist countries, land use is subject to planning approval, and central urban areas usually have good public facilities, such as large parks, stadiums, exhibition halls, and other public service facilities, which can ensure the sustainable demographic attractiveness of central urban areas. Most cities in the Beijing–Tianjin–Hebei region showed the characteristics of continuous agglomeration in central urban areas. However, due to great differences in the development levels among cities in the Beijing–Tianjin–Hebei region, agglomeration characteristics and city dispersion are different. Population changes in metropolitan areas in European and North American countries still show different characteristics. Central metropolitan areas in European and North American countries exhibit dispersion and rejuvenation characteristics, which are closely related to the urban development stage [61]. However, the reasons for the population change in the Beijing–Tianjin–Hebei region suggest a strong policy orientation. For example, the policy to relieve Beijing of functions nonessential to its role as China's capital was released in 2014. From 2000 to 2010, Beijing showed global growth characteristics. However, from 2010 to 2017, Beijing's inner-city population decreased. Although the population in peripheral areas maintained growth, the intensity also decreased. Accordingly, populations in other areas of Beijing, Tianjin, and Hebei obtained development opportunities. Through the comparison of the two, the population distribution and change in cities were closely related to the urban development stage and process. The influence of the urban market on capitalist countries is significant, whereas cities in socialist countries are under the dual control of government and market.

*4.2. Urban and Rural Coordinated Development*

Rural areas in European and American countries have been gradually undergoing transformation and development under the impact of external forces. A large number of immigrants entered the countryside, and the functions of the countryside gradually diversified [63]. However, due to the different stages of development, relatively large differences between the urban and rural areas in China were observed. According to Todaro's theory of urban–rural population mobility [64], developing countries should control the continuous expansion of urban–rural differences. If the differences between urban and rural areas are too large, then many people are concentrated in urban areas, causing many urban problems as well as labor shortages in rural areas. The new type of urbanization planning, issued in 2014, (http://www.gov.cn/zhengce/2014-03/16/content_2640075.htm, accessed on 1 January 2021) also emphasized urban and rural development integration. The plan proposed to enhance the vitality of rural development, gradually narrow the gap between urban and rural areas, and promote the coordinated promotion of urbanization and new rural construction.

From the regional development perspective, urban–rural integration should not only balance the differences within administrative divisions but also allocate population and other production factors from the macro perspective of urban agglomeration. The integrated development of urban agglomeration emphasizes integration among cities and urban and rural areas; the integration of urban and rural areas is the basis of urban

agglomeration [30]. From the above analysis, the Beijing–Tianjin–Hebei region population distribution imbalance gradually increased, and urban and rural differences gradually became obvious. Two major trends existed in population agglomeration in the Beijing–Tianjin–Hebei region: (1) continuous agglomeration to Beijing and Tianjin at the regional scale; (2) within a county, the population gathered in the township unit where the county government was located.

Population is an important factor in the Beijing–Tianjin–Hebei coordinated development process. At the regional scale, excessive population agglomeration to megacities is not conducive to urban development integration. We should cultivate satellite cities with improved traffic conditions and infrastructure while relieving Beijing of functions nonessential to its role as China's capital in an orderly fashion. At the township level, the effects of units where administrative centers were located and key towns on population growth in surrounding areas were not obvious. In the future, the industrial and economic links between administrative units and surrounding areas must be strengthened and the regional growth pole (key towns) should be cultivated to promote the development of surrounding areas using their diffusion effect.

### 4.3. Significance and Shortcomings of Township-Level Research

In this study, population distribution and change characteristics were explored at the county scale in the Beijing–Tianjin–Hebei region. Township-level data provided three main advantages. First, the population distribution and change characteristics within district and county units were explored on the basis of the population data at the township level. Existing studies only described trends of population concentration and dispersion in the Beijing–Tianjin–Hebei region at the city scale [24]. However, under the background of rapid urbanization, the population changes within the county were also fierce and worthy of attention. Second, the effects of policy factors at the township level were analyzed, such as the setting of a key town in the planning that does not promote the sustained growth of the population of the town—which remains to be further cultivated. Most of the existing policy studies were based on cities [65]. However, this paper provided some support for the policy evaluation of key towns. Last, we were able to understand the development of urban–rural differences from a microscopic scale, and such differences could then be described accurately at the county scale. Due to the availability of township-scale data, a distinction between natural and mechanical changes in population was lacking. In the analysis of influencing factors, more accurate data of rural socioeconomic factors would be needed to further explore diverse influencing factors.

### 5. Conclusions

Taking a typical urbanizing megaregion of China, the Beijing–Tianjin–Hebei region, as an example, and drawing on the method of population concentration index, population density classification, and multivariable linear regression model, this paper depicted population dynamics and explored influencing factors at the township-level. The main conclusions drawn were as follows:

First, the population distribution in the Beijing–Tianjin–Hebei region was uneven, as reflected in an observed east–middle–west pattern in the regional scale and a center-periphery pattern in the county scale. Additionally, the coordinated development strategy promoted the decentralization of population, albeit to a limited extent. During 2010–2017, the population continued to concentrate in the municipal districts of Beijing and Tianjin and township units where county governments were located, which caused a more unbalanced population geographical distribution. However, compared with the period of 2000–2010, the speed of population concentration slowed down slightly in 2010–2017, and some townships in the urban area of Beijing lost population, indicating that the coordinated development strategy of Beijing–Tianjin–Hebei region promoted decentralization of population, albeit to a limited extent.

Second, population changes in townships within a county were diversely affected by the location of the township and the development of the county. Most of the townships where the county governments were located experienced continuous population growth since 2000. Other townships, however, exhibited differentiated population dynamics, depending on the development of the county. In undeveloped counties, population loss was observed in the suburban and remote area of townships. With the development of the county to a certain stage, townships in suburbs began to attract population as industries moved in from central cities. As the county developed further, some suburban townships lost population while some remote townships became attractive, similar to what Beijing has experienced.

Finally, although the population changes were influenced by the market and the government, the population distribution was constrained by natural factors to a large extent. The influence of the government on population changes was mainly reflected in two aspects. First, townships where the county governments were located experienced faster population growth than other regions. Second, owing to the strategy of decentralizing the non-capital functions of Beijing, the trend of population concentration in Beijing slowed down to a certain extent, and areas outside Beijing gained stronger population attraction. However, not all policies worked. The influence of key towns on population changes was not evident, suggesting that additional works will be needed to support the development of key townships and enhance their attractiveness to people.

Our future work will mainly focus on two aspects: first, we will try to identify the leading factors affecting population changes in Township units. Because there are differences in the leading factors of population changes in different types of township units, we need to analyze and identify those specific factors to better provide different population guidance policies. On the other hand, it is very important to establish a database based on different data sources of township units to provide data support for township research.

**Author Contributions:** Conceptualization, T.L. and Y.W.; methodology, Y.W. and T.L.; formal analysis, Y.W. and Y.Z.; resources, T.L.; data curation, Y.W.; writing—original draft preparation, Y.W. and T.L.; writing—review and editing, Y.W. and T.L.; visualization, Y.W. and T.L.; supervision, T.L. and Y.Z.; project administration, T.L.; funding acquisition, T.L. All authors have read and agreed to the published version of the manuscript.

**Funding:** This research was funded by the National Key Research and Development Program of China (2019YFD1100803; 2018YFD1100803).

**Institutional Review Board Statement:** Not applicable.

**Informed Consent Statement:** Not applicable.

**Data Availability Statement:** The census data for 2000–2010 are publicly available from the National Bureau of Statistics of China (http://www.stats.gov.cn/, accessed on 1 January 2021). The 2020 census needs to be obtained by consulting the Seventh Census Bulletins issued by local governments in China, which can be found online at: https://tjgb.hongheiku.com/%e4%b8%ad%e5%9b%bd, accessed on 1 January 2021. The "China County Statistical Yearbook (Township Volume) -2018" can be obtained through the China National Knowledge Infrastructure (https://data.cnki.net/yearbook/Single/N2019070153, accessed on 1 January 2021). The "Beijing Statistical Yearbook 2018", the "Tianjin Statistical Yearbook 2018" and the "Hebei Economic Yearbook 2018" are available at: http://nj.tjj.beijing.gov.cn/nj/main/2018-tjnj/zk/indexch.htm, accessed on 1 January 2021; http://stats.tj.gov.cn/nianjian/2017nj/zk/indexch.htm, accessed on 1 January 2021; http://tjj.hebei.gov.cn/res/nj2018/zk/indexch.htm, accessed on 1 January 2021.

**Conflicts of Interest:** The authors declare no conflict of interest.

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
