# Peer review of "Population Dynamics in China’s Urbanizing Megaregion: A Township-Level Analysis of the Beijing–Tianjin–Hebei Region"

_land, doi:10.3390/land11091394_

Round 1
Reviewer 1 Report
This article appears to be well organized and structured. It uses appropriate methodologies to explore the evolution of the spatial pattern of population density and the influencing factors in the Beijing–Tianjin–Hebei region at the township level. I didn't notice any errors in the text, figures, or tables. However, some mistakes need to be corrected: in the Conclusion, the last paragraph in lines 745 to 754 is duplicated.
Reviewer 2 Report
The manuscript titled " Population Dynamics in China’s Urbanizing Megaregion: A Township‐level Analysis of the Beijing–Tianjin–Hebei Region " intends to explore the population distribution and change characteristics of the Beijing–Tianjin–Hebei region at the township level, based on permanent population data in 2000, 2010, and 2017. The analysis used the population concentration index, which is a quantitative index to judge the concentration of regional population distribution.
The research is original; it could be characterized as novel and in my opinion important to the field, it also has an almost appropriate structure, and the language has been used well. In the meanwhile, the manuscript has a big extent (about 9,536 words) and it is comprehensive. The tables (3) and figures (10) make the paper reflect well to the reader. For this reason, paper has a "diversity look", not only tables, not only numbers, not only words. It is advised to revise figures, compare them, or use appendix if you agree.
The title, I think, is all right. The abstract did not reflect well the findings of this study, and it was not the appropriate length. Please revise the abstract of the manuscript and do not forget abstract need to encourage readers to download the paper. The Abstract needs further work. It is not clear. Abstracts should indicate the research problem/purpose of the research, provide some indication of the design/methodology/approach taken, the findings of the research and its originality/value in terms of its contribution to the international literature. The abstract has a long length (about 276 words). Please, revise the abstract, it must be up to 200 words long, for this reason I would be good to reduce [see: Instructions for Authors / Manuscript Submission Overview / Accepted File Formats - (https://www.mdpi.com/journal/land/instructions#submission or https://www.mdpi.com/files/word-templates/land-template.dot)].
Please, revise the line 173 and make the appropriate currency exchange using dollars ($) or euros (€) or both also you can keep yuan. This is because the results of the research must be directly comparable to other similar surveys that have already been carried out around the world and other such surveys will certainly be carried out, and do not forget, the journal “Land” is international.
The introduction is effective, clear, and well organized but it wasn’t introduced and put into perspective what research is negotiating. Moreover, it does not contain a clear formulation and description of the research problem. Please insert a clear description and justification of the problem the article deals with. Your literature research should be critical and more informed, rather than listing previous research. This section requires significant improvement.
For the Methodology chapter, the research conduct has been tested in several areas of the world, with comparable results and will probably be tested in others. Appropriate references to the methodology included in the already published bibliography but you can put more references, from all over the world. Do not forget, the journal “Land” is international.
The results section is good. The argument flows and is reinforced through the justification of the way elements are interpreted. But the same does not apply to the Discussion and Conclusion. It is advised to revise the Discussion and Conclusion. Both sections should be consistent in terms of Proposal, Problem statement, Results, and of course, future work. Your conclusion section does not do justice to your work. Make your key contributions, arguments, and findings clearer. You must refer to the literature and previous studies in your discussion section.
Please revise the manuscript and include more references which already exist in the bibliography. I would be much more satisfied if the number of references was slightly higher (about 15 - 20 references) and I would appreciate it if it also included data other than Asia for example America, Europe, or Australia, because it has many references from Asia (about 20 in Chinese). In this way it is documented that a method that is tested in a place with its own characteristics can be implemented in other places around the world.
More discussion is needed, comparing the results of this work related to attributes with those of other studies. I believe that the conclusions section or discussion should also include the main limitations of this study and incorporate possible policy implications. I think, something more should be said about practical implications.
Reviewer 3 Report
Remarks and questions to be answered:
- the relevant weakness is paper could not be interesting for international readers,
- please explain in detail the criteria for the selection of study area,
- why have you selected those variables?
- what are the research questions in this paper?
- conclusions are somewhat descriptive, and no valuable recommendations for socio-economic practice, which is a severe flaw.
Round 2
Reviewer 3 Report
Accept in present form.